# Longitudinal Dynamics of Immune Response in Occupational Populations Post COVID-19 Infection in the Changning District of Shanghai, China

**DOI:** 10.3390/v16050672

**Published:** 2024-04-25

**Authors:** Li Li, Fengge Wang, Xiaoding He, Tingting Pei, Jiani Lu, Zhan Zhang, Ping Zhao, Jiayu Xue, Lin Zhu, Xinxin Chen, Zijie Yan, Yihan Lu, Jianlin Zhuang

**Affiliations:** 1Shanghai Changning Center for Disease Control and Prevention, 39 Yunwushan Road, Shanghai 200051, China; 18116019796@163.com (L.L.); hexiaoding@cncdc.org (X.H.); ptt_1988@126.com (T.P.); lujiani@cncdc.org (J.L.); zhangzhan@cncdc.org (Z.Z.); zhaoping@cncdc.org (P.Z.); xuejiayu@cncdc.org (J.X.); zhulin@cncdc.org (L.Z.); chen_xinxin1228@163.com (X.C.); yanzijie960213@foxmail.com (Z.Y.); 2Shanghai Institute of Infectious Disease and Biosecurity, Fudan University, 130 Dong’an Road, Shanghai 200032, China; fgwang22@m.fudan.edu.cn; 3Department of Epidemiology, Key Laboratory of Public Health Safety of Ministry of Education, School of Public Health, Fudan University, 130 Dong’an Road, Shanghai 200032, China

**Keywords:** SARS-CoV-2, antibody, cellular immunity, occupational population, dynamic changes

## Abstract

Monitoring the long-term changes in antibody and cellular immunity following Severe Acute Respiratory Syndrome Coronavirus 2 (SARS-CoV-2) infection is crucial for understanding immune mechanisms that prevent reinfection. In March 2023, we recruited 167 participants from the Changning District, Shanghai, China. A subset of 66 participants that were infected between November 2022 and January 2023 was selected for longitudinal follow-up. The study aimed to investigate the dynamics of the immune response, including neutralizing antibodies (NAbs), anti-spike (S)-immunoglobulin G (IgG), anti-S-IgM, and lymphocyte profiles, by analyzing peripheral blood samples collected three to seven months post infection. A gradual decrease in NAbs and IgG levels were observed from three to seven months post infection. No significant differences in NAbs and IgG titers were found across various demographics, including age, sex, occupation, and symptomatic presentation, across five follow-up assessments. Additionally, a strong correlation between NAbs and IgG levels was identified. Lymphocyte profiles showed a slight change at five months but had returned to baseline levels by seven months post infection. Notably, healthcare workers exhibited lower B-cell levels compared to police officers. Our study demonstrated that the immune response to SARS-CoV-2 infection persisted for at least seven months. Similar patterns in the dynamics of antibody responses and cellular immunity were observed throughout this period.

## 1. Introduction

Since 2022, the Omicron variant of Severe Acute Respiratory Syndrome Coronavirus 2 (SARS-CoV-2) has become the dominant strain, exhibiting increased transmissibility and immune evasion, along with a reduction in virulence [1,2,3,4]. Our understanding of immune responses to SARS-CoV-2 remains limited. Naturally, people who are infected with COVID-19 generate different types of antibodies: immunoglobulin M (IgM), immunoglobulin G (IgG), and immunoglobulin A (IgA) [5]. IgG is the most common type of antibody in serum. Anti-spike (S)-IgG refers to IgG antibodies that target the S protein. Since the receptor-binding domain (RBD) of the S protein binds to human ACE2 receptor, antibodies targeting this domain are likely to neutralize SARS-CoV-2 [6,7,8,9]. Thus, they are known as neutralizing antibodies (NAbs). Monitoring NAbs levels is vital for assessing immune status and resistance to COVID-19. Anti-nucleocapsid (N) protein-IgG refers to IgG antibodies targeting the N protein. While these antibodies can help manage viral infections by activating other parts of the immune system, they typically lack neutralization activity [10]. IgM antibodies typically appear within 3–7 days post symptom onset (PSO) and diminish in the later infection stages. IgG antibodies, which emerge approximately 7–14 days post infection, persist into the convalescent phase [11,12]. Thus, IgM and IgG antibodies signify acute-phase infection and either past infection or convalescence, respectively [13]. Nearly all individuals infected with SARS-CoV-2 develop seropositive antibodies within three weeks PSO [14]. Clinically, patients with COVID-19, especially those with severe symptoms, exhibit marked lymphocyte depletion, neutrophil elevation, and cytokine accumulation [15].

Although the Coronavirus Disease 2019 (COVID-19) pandemic trend is slowly flattening, acquiring further knowledge on the magnitude, timing, and longevity of antibody responses following SARS-CoV-2 infection remains essential for understanding the role of antibodies to immunity. Dysregulation of the immune response occurs at the onset of COVID-19. However, the long-term immunological dynamics associated with COVID-19 remain incompletely understood. Epidemiological studies have reported a rapid decline in humoral immunity within four to ten weeks post infection, while others have indicated that antibodies can persist for up to six months or longer [16,17,18,19,20]. Furthermore, a significant study reported a moderate decline in antibodies targeting the SARS-CoV-2 S protein and RBD over an eight-month period, while specific T-cell-mediated immune responses remained detectable at eight months post infection [17].

Conducted in the Changning District of Shanghai, this study dynamically monitored SARS-CoV-2-specific humoral and cellular immunity, spanning from three to seven months following the nationwide outbreak of COVID-19 in December 2022 in China. It focused on professional cohorts with the aim of assessing longitudinal changes in antibody levels following natural infection and exploring their correlation with individual characteristics, thereby providing insights into the interaction patterns between the virus and the human body.

## 2. Materials and Methods

### 2.1. Study Design and Participants

According to the “New Coronavirus Pneumonia Prevention and Control Protocol for COVID-19 (10th Edition)” [21], a confirmed COVID-19 case is defined as an individual with a relevant epidemiological history and clinical symptoms, confirmed by a positive SARS-CoV-2 nucleic acid test via RT-PCR. All participants in this study were informed about the study and signed an informed consent form. Exclusion criteria include (1) individuals who refused to participate, (2) those with severe acute or chronic diseases, or in the active phase of a chronic disease, including autoimmune diseases or tumor diseases, and (3) those whose consecutive nucleic acid tests for SARS-CoV-2 (using real-time PCR with a threshold of 40) showed cycle threshold (Ct) values of 35 or lower, with samples taken at least 24 h apart. In this cohort study, conducted in the Changning District, 167 participants were recruited from November 2021 to January 2023 during the COVID-19 pandemic. Among them, 114 were diagnosed with COVID-19, with two distinct infection periods observed: 13 cases occurred between November 2021 and May 2022, and 101 cases from November 2022 to January 2023 (Figure 1). In March 2023, we conducted the first follow-up of all participants through questionnaire surveys and immune response tests. To further investigate the dynamics of the post-recovery immune response, a comprehensive follow-up was carried out on 66 individuals infected between November 2022 and January 2023, conducted at four different times. Peripheral blood samples for antibody assessment were longitudinally collected in April, May, June, and July 2023, and cellular immunity tests were conducted in May and July 2023.

### 2.2. Laboratory Examination and Data Collection

Demographic data, including sex, age, occupation, vaccination status, medical history, date of diagnosis, number of vaccine doses received, and date of last vaccination, were collected. The SARS-CoV-2 IgM, IgG, and NAbs were detected using a magnetic particle chemiluminescence method with kits from Zhengzhou Antu Bioengineering Co., Ltd. (Zhengzhou, China). Samples from each participant were analyzed for the levels of two specific anti-SARS-CoV-2 antibodies targeting the S protein (anti-S-IgG and anti-S-IgM). The levels of IgM and IgG were quantified as the luminescence value/cut-off value (S/CO), with an S/CO ≥ 1.0 considered positive and <1.0 considered negative. NAbs were assessed using the receiver operating characteristic (ROC) curve method, with levels < 30 AU/mL defined as negative and ≥30 AU/mL as positive. Cellular immune indicators were detected using flow cytometry, provided by Beckman Coulter Co., Ltd. (Brea, CA, USA). In particular, CD3-CD19+ was used as the marker for B cells, and CD3-CD16+CD56+ for NK cells, accurately distinguishing each cell type with specific marker combinations. All sample analyses and result interpretations were conducted in accordance with the instructions.

### 2.3. Statistical Analysis

Data were represented as mean ± standard deviation (SD) for normally distributed variables and as median with interquartile range (IQR) for non-normally distributed variables. The titers of IgG and NAbs were log-transformed (base 10) for geometric mean calculations. Analysis was conducted using R software (version 4.3.1). The Mann–Whitney U test and chi-squared tests were used to compare continuous variables and proportions, respectively. A linear mixed-effects model was employed to assess the impacts of sex, age, occupation, symptomatic status, and underlying diseases across five follow-up periods. Statistical significance was set at *p* < 0.05.

## 3. Results

### 3.1. Baseline Characteristics of Participants

In this survey, 167 participants were enrolled, including 97 males (58.08%) and 70 females (41.92%), with an average age of 41.28 years. The cohort included 50 police officers (29.94%), 54 healthcare workers (32.34%), and 63 community residents (37.72%) (Table 1). Of these, 161 participants (96.41%) were vaccinated: one participant (0.60%) received a single dose, 16 participants (9.58%) received two doses, 117 participants (70.06%) received a booster for a total of three doses, and 27 participants (16.16%) received two boosters, totaling four doses, primarily administered between December 2022 and January 2023 (Table 1).

NAbs levels were significantly higher in previously infected participants (1259.36 ± 1039.63 AU/mL) than in uninfected ones (530.80 ± 622.33 AU/mL) (*p* = 0.013), indicating a robust humoral response post infection. A high seroprevalence of antibodies was observed in two groups (Table 1). In this cohort of 167 participants, 114 (68.26%) tested positive for SARS-CoV-2, while 53 (31.74%) tested negative. IgG antibodies were detected in both groups, suggesting background immunity or cross-reactivity. IgM antibodies, markers of recent infection, showed low prevalence in infected participants, likely due to the timing of testing occurring three months post infection. Overall, a significant correlation was observed between previous infection and increased antibody levels.

In this study, 114 participants (68.26%) reported previous SARS-CoV-2 infections, including 13 cases from November 2021 to May 2022 and 101 cases from November 2022 to January 2023 (Table 2). Among the unvaccinated group (6 participants), all reported previous infections, whereas in the vaccinated group (108 participants), the infection rate was 67.08%. Of the 114 infected participants, 4 people tested positive for IgM, with a positivity rate of 3.51%. All 114 participants tested positive for IgG, with an average of 90.10 ± 40.77 S/CO). Moreover, 110 participants tested positive for NAbs, with a positivity rate of 96.49% and an average of 1259.36 ± 1039.63 AU/mL (Table 2).

### 3.2. Dynamic Changes in Antibody Levels across Five Follow-Up Visits

Of the 101 participants infected between 15 November 2022, and 15 January 2023, 66 occupational participants were followed up between March and July 2023, providing five sets of antibody measurements and three assessments of immune indicators. This subgroup included 45 males and 21 females, aged between 24 and 55 years, with an average age of 35.98 years. The cohort included 32 healthcare workers and 34 police officers. Of these, 24 participants presented symptoms such as cough and fever, while 42 were asymptomatic. Ten participants had chronic conditions, primarily hypertension and diabetes. All had received at least two doses of COVID-19 vaccine.

In a cohort of 66 patients, log-transformed SARS-CoV-2 NAbs titers were monitored, with median (IQR) values at 3.39 (3.17, 3.57), 3.29 (3.06, 3.53), 3.28 (2.99, 3.50), 3.27 (3.05, 3.48), and 3.23 (3.03, 3.58) for the three to seven months post-infection follow-ups, respectively (Appendix A). Trends in antibody levels across different demographic groups, such as sex, age, occupation, symptomatic status, and the presence of underlying diseases, were investigated using a repeated measures linear mixed-effects model. Significant differences in NAbs titers were observed between the five follow-ups. However, no significant impact of these demographic factors on antibody levels was found (Figure 2 and Table 3).

In our follow-up studies, 18 participants exhibited increasing NAbs levels at five, six, and seven months post infection (Appendix A). Although these participants did not report any reinfections during the questionnaire survey, the observed increase in antibody levels could suggest a possible reinfection with SARS-CoV-2. Interestingly, this corresponds with our findings during antibody testing, where a few participants tested positive for IgM antibodies. However, given the typically short-lived nature of IgM antibodies, which served as the early immune response and which are generally detectable within a few days after the onset of COVID-19 symptoms, IgM antibodies were not detectable in COVID-19 cases across the majority of the five follow-up assessments in our study.

The COVID-19 IgG antibody levels demonstrated a gradual decline over time, with median (IQR) values of 2.13 (1.92, 2.35), 2.04 (1.82, 2.28), 2.04 (1.81, 2.26), 1.98 (1.79, 2.21), and 1.98 (1.72, 2.18) at three to seven months post infection (Appendix A). Variations in IgG antibody levels were analyzed considering sex, age, occupation, symptomatic status, and underlying health conditions using a repeated measures linear mixed-effects model. This analysis revealed a significant downward trend in IgG antibody levels from the first follow-up post infection. However, no significant statistical impact of demographic factors on IgG levels was observed (Figure 3 and Table 3).

Our follow-up testing revealed a significant positive correlation between NAbs and IgG levels throughout five follow-up assessments post infection (Appendix A). This was attributable to the fact that the antibody detection kits used in this study target the S protein of the virus. These anti-S-IgG likely possessed viral neutralizing activity. Furthermore, the increasing correlation suggested a persistent enhancement and integration of the antibody response over time.

### 3.3. Kinetic Analysis of Lymphocyte Profile across Follow-Up Visits

To assess the kinetic changes in various lymphocyte subsets in the peripheral blood of COVID-19 patients over three follow-up visits, flow cytometry was used to analyze subsets of CD3+ T cells, CD4+ T cells, CD8+ T cells, B cells, and NK cells. Notably, significant changes in the percentages and absolute counts per microliter (µL) of CD4+ T cells, CD8+ T cells, B cells, and NK cells were observed between three and five months. However, by seven months, these changes showed no difference compared to levels observed at three months post infection (Table 4 and Figure 4). The percentage and absolute counts/µL of B cells among healthcare workers were lower than those in police officers. In contrast, other immune markers, such as the percentage of CD3+ T cells and the CD4+/CD8+ ratio, did not exhibit notable changes during the follow-ups in most demographics, suggesting relatively stable cellular immunity across different populations. Lymphocyte levels remained generally stable and within normal ranges. Although the abnormality rate of the CD3+ lymphocyte percentage increased slightly, other subsets exhibited decreased or unchanged abnormality rates from three to seven months (Appendix A).

## 4. Discussion

Understanding the duration and stability of the immune response, including immunophenotyping and antibodies, in the COVID-19 recovery period is essential for predicting protective immunity and interpreting serological and epidemiological data. This study demonstrated that, despite a declining trend in antibody levels, there remains a relatively high level of positivity seven months after infection, aligning with existing publications [17,22]. The results of NAbs and IgG were consistent. Furthermore, other lymphocyte subset levels remained stable, with a generally decreasing rate of abnormalities.

Recent studies on long-term immunity in individuals recovered from COVID-19 indicate that immunological memory leads to the production of specific antibodies against SARS-CoV-2. Initially, antibody levels rapidly increase post infection, and then decrease and stabilize as the immune system controls and eliminates the virus, demonstrating dynamic immunological balance [23]. A 2020 study in China found that 80% of patients recovering from COVID-19 retained antibodies up to a month post recovery [24]. Further studies on antibody persistence have shown various durations, ranging from six to fifteen months post infection [25,26,27]. In this study, levels of NAbs and IgG antibodies remained relatively high in follow-ups from three to seven months after the widespread infections in China in December 2022, although they showed a gradual decline. Notably, there was a significant positive correlation between NAbs and IgG antibodies (*p* < 0.05). Several participants experienced an increase in NAbs between five and seven months, potentially due to reinfection. Few IgM antibodies were detected during the follow-ups, consistent with the known role in early viral defense. A longitudinal study conducted in Beijing reported that 85.71% (48/56) of IgM-positive patients transitioned to IgM seronegative status, some as early as 32 days PSO [28]. 

A linear mixed-effects model was used to assess the impact of age, underlying diseases, and vaccination status on antibody levels. The analysis revealed no significant differences in NAbs levels across demographic groups, likely due to the homogeneity of the sample, which consisted of 66 young, healthy professionals with mild or asymptomatic COVID-19. However, other studies demonstrated significant differences in NAbs levels between patients with mild/moderate symptoms and those with severe symptoms, suggesting a correlation between the antibody response and disease severity [22]. Regarding the IgG response to SARS-CoV-2, a longitudinal study observed a more rapid decrease in anti-N IgG titers among younger and asymptomatic individuals [29]. Another study reported that the anti-N-IgG S/CO index declined over time, with half-lives of 75.4 days in asymptomatic individuals and 107.6 days in pneumonia cases [30]. This pattern potentially reflected greater viral exposure or interaction in patients with more severe symptoms. However, in our study, IgG levels did not exhibit significant differences across various demographic groups at the evaluated time points. Additionally, we utilized various models to simulate antibody change trends, but these models aligned only partially with the data. This inconsistency might be attributed to COVID-19 reinfections among some participants, causing abrupt fluctuations in antibody levels. Moreover, the prolonged impacts of long-COVID undoubtedly affect many patients over time [31]. Thus, the trends in immune response and associated risk factors require ongoing long-term monitoring.

Lymphocytes, which are categorized into T cells, B cells, and NK cells, play crucial roles in immune regulation. During COVID-19 infection, peripheral blood levels of CD3+ T cells, CD4+ T cells, CD8+ T cells, B cells, and NK cells notably decrease. The reduction is likely due to the recruitment of these cells from the bloodstream to tissues and organs in response to SARS-CoV-2 infection [32,33]. This study found that the lymphocyte profile slightly changed at 5 months and then returned to baseline by 7 months post infection. This fluctuation may be attributed to reinfections in some patients, consistent with findings on the trend in NAbs. While our study indicated that the median (IQR) values of lymphocyte subsets, including T cells, B cells, and NK cells, remained within normal ranges during follow-ups, we observed a relatively high rate of abnormalities, particularly in the absolute counts of B and NK lymphocytes. CD8+ effector T cells might undergo redistribution and exhaustion in response to SARS-CoV-2, potentially delaying the normalization of the immune system [34]. Current studies indicate that most COVID-19 patients maintain strong antibody and T-cell immunity against the virus in a long-term period [17,35,36,37]. In our study, healthcare workers exhibited significant lower levels in both the percentage and count/µL of B lymphocytes compared to police officers in the follow-ups. However, the underlying mechanisms remained unknown. Although evidence has suggested a correlation between prior lung conditions and increased CD8+ T cell responses specific to SARS-CoV-2 [36], our findings did not show a direct association between lymphocyte profiles and underlying diseases post-COVID-19. Another study observed the elevated activation of SARS-CoV-2-specific T cells in patients recovered from COVID-19 compared to healthy participants [38]. However, our study did not conduct such comparisons.

This study has several limitations. First, the cohort predominantly consisted of cases ranging from asymptomatic to moderate COVID-19, with difficulties analyzing the relationship between immune response and disease severity. Second, the sample size was limited during the post-SARS-CoV-2 infection phase. Third, since the monitoring of antibody dynamics commenced only from the third month post infection, it was not possible to accurately determine the peak levels of IgM and IgG during the acute phase. Additionally, the absence of pre-infection baseline data for NAbs and IgG antibodies posed challenges for the assessment of antibody levels that might influence susceptibility to COVID-19. Furthermore, the absence of investigation into adaptive cell-mediated responses and specific subpopulations of T and B cells limited our ability to thoroughly explore the state of cellular immunity. Despite these limitations, our study contributed to the understanding of the immune responses to COVID-19, with potential implications for vaccine development. 

## 5. Conclusions

This study explored the antibody and cellular immune responses following SARS-CoV-2 infection. Our findings revealed that both NAbs and IgG levels remained relatively high up to seven months post infection. We observed similar decline trends in NAbs and IgG levels across different demographic groups, with a notable correlation between these indicators. The lymphocyte profile decreased slightly at five months post infection, and then returned to baseline by seven months. Healthcare workers showed lower B cell levels than police officers. Most lymphocyte subsets demonstrated a gradual decline in abnormality rates, although B cell and NK cell counts showed higher rates of abnormalities. Future studies should employ detailed cellular markers and functional assays to expand on the preliminary findings and incorporate larger sample sizes with comprehensive and extended follow-up to thoroughly understand the dynamics and longevity of immunity post infection.

## Figures and Tables

**Figure 1 viruses-16-00672-f001:**
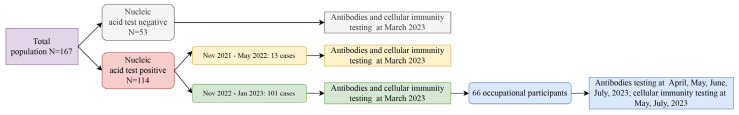
Flow chart of a population cohort: longitudinal follow-up study on COVID-19 infections.

**Figure 2 viruses-16-00672-f002:**
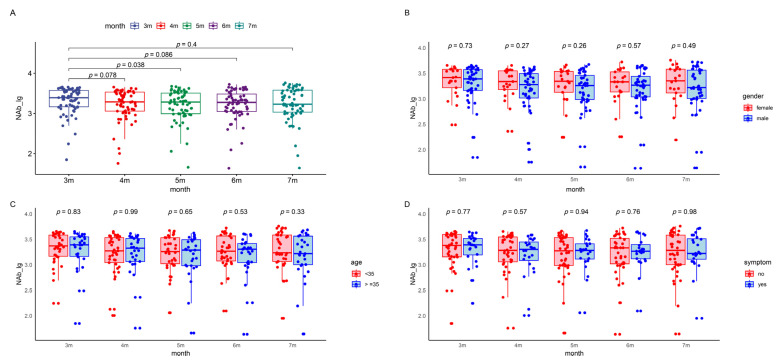
NAbs levels in the patients during the five follow-up visits. The boxes represent the distribution of the NAbs levels in the patient population for different visits. (**A**) Total NAbs level changes over time. (**B**) Gender-based comparison between male and female participants. (**C**) Age-based comparison of participants younger than 35 with those 35 and above. (**D**) Symptom-based comparison between symptomatic and asymptomatic patients. (**E**) Occupation-based comparison between healthcare workers and police officers. (**F**) Underlying-disease-based comparison between participants with and without underlying diseases. Lg, logarithmic value to the base 10. NAb, neutralizing antibody.

**Figure 3 viruses-16-00672-f003:**
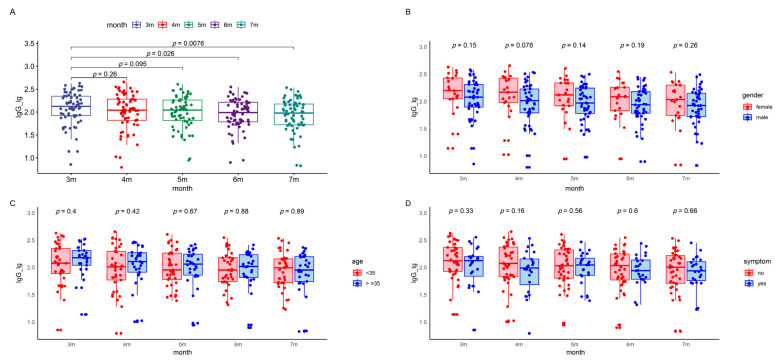
IgG levels in the patients during the five follow-up visits. The boxes represent the distribution of the IgG levels in the patient population for different visits. (**A**) Total IgG level changes over time. (**B**) Gender-based comparison between male and female participants. (**C**) Age-based comparison of participants younger than 35 to those 35 and above. (**D**) Symptom-based comparison between symptomatic and asymptomatic patients. (**E**) Occupation-based comparison between healthcare workers and police officers. (**F**) Underlying disease-based comparison between participants with and without underlying diseases. Lg, logarithmic value to the base 10. Lg, logarithmic value to the base 10. IgG, immunoglobulin G.

**Figure 4 viruses-16-00672-f004:**
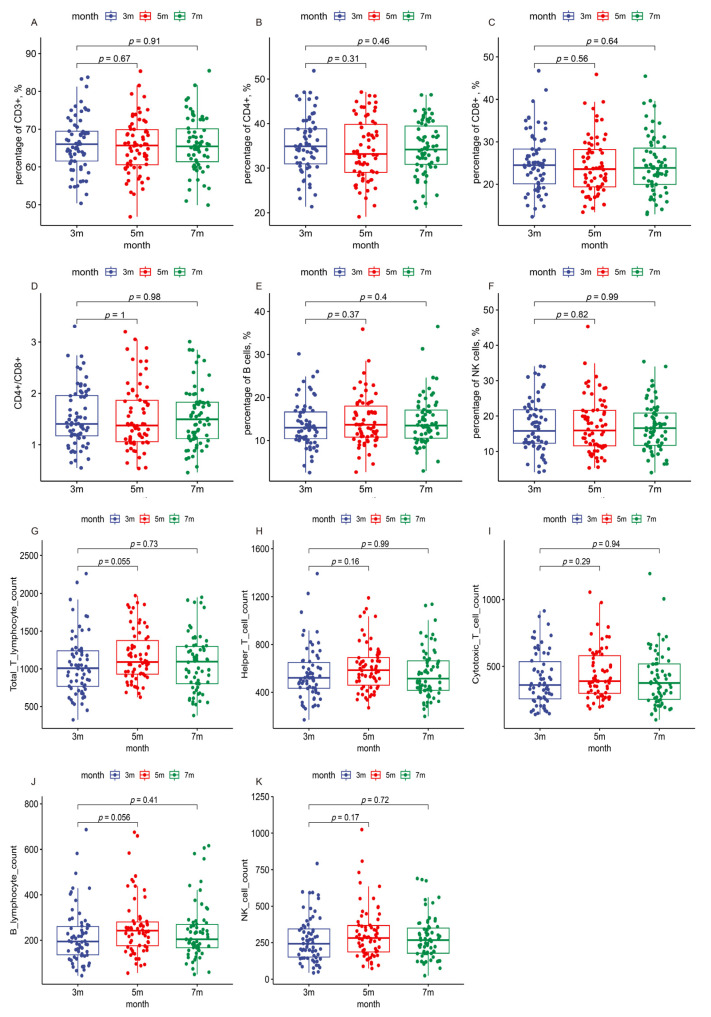
Immune indicators in the patients during the three follow-up visits. The boxes represent the distribution of the immune cell levels in the patient population for different visits. (**A**) Percentage of CD3+ T cells; (**B**) percentage of CD4+ T cells; (**C**) percentage of CD8+ T cells; (**D**) CD4+/CD8+ ratio; (**E**) percentage of B cells; (**F**) percentage of NK cells; (**G**) absolute count of total T cells per microliter; (**H**) absolute count of helper T cells per microliter; (**I**) absolute count of cytotoxic T cells per microliter; (**J**) absolute count of B cells per microliter; (**K**) absolute count of NK cells per microliter.

**Table 1 viruses-16-00672-t001:** Baseline characteristics and antibody responses of participants in the Changning District of Shanghai.

	Total	Infected	Uninfected	*p*
N	167	114 (68.26)	53 (31.74)	
Age, X ± SD	41.28 ± 14.96	41.70 ± 15.08	40.38 ± 14.80	0.596
Age group				0.137
0–19	7 (4.19)	3 (2.63)	4 (7.55)	
20–39	70 (41.92)	51 (44,74)	19 (35.85)	
40–59	62 (37.13)	38 (33.33)	24 (45.28)	
60–79	28 (16.77)	22 (19.30)	6 (11.32)	
Sex				0.810
Male	97 (58.08)	65 (57.02)	32 (60.38)	
Female	70 (41.92)	49 (42.98)	21 (39.62)	
Underlying disease				0.730
No	128 (76.65)	86 (75.44)	42 (79.25)	
Yes	39 (23.35)	28 (24.56)	11 (20.75)	
Vaccine dose(s)				**0.016**
No	6 (3.59)	6 (5.26)	0	
One	1 (0.60)	1 (0.88)	0	
Two	16 (9.58)	8 (7.02)	8 (15.09)	
Three	117 (70.06)	86 (75.44)	31 (58.49)	
Four	27 (16.17)	13 (11.40)	14 (26.41)	
Occupation				0.204
Police officer	50 (29.94)	39 (34.21)	11 (20.75)	
Healthcare worker	54 (32.34)	34 (29.82)	20 (37.74)	
Community populations	63 (37.72)	41 (35.96)	22 (41.51)	
NAb ^1^				
GM ± GSD	957.34 ± 920.84	1259.36 ± 1039.63	530.80 ± 622.33	**0.013**
Negative	8 (4.79)	4 (3.51)	4 (7.55)	
Positive	159 (95.21)	110 (96.49)	49 (92.45)	
IgG				
GM ± GSD	79.04 ± 43.40	90.10 ± 40.77	59.62 ± 41.97	0.380
Negative	0	0	0	
Positive	167 (100)	114 (100)	53 (100)	
IgM				/
Negative	159 (95.21)	110 (96.49)	49 (92.45)	
Positive	8 (4.79)	4 (3.51)	4 (7.55)	

^1^ GM, geometric mean; GSD, geometric standard deviation; NAb, neutralizing antibody; IgG, immunoglobulin G; IgM, immunoglobulin M. Values in bold indicate statistical significance (*p* < 0.05).

**Table 2 viruses-16-00672-t002:** Baseline characteristics of participants infected with SARS-CoV-2 between November 2021 and January 2023 in the Changning District of Shanghai.

	Total	November 2021–May 2022	November 2022–January 2023
N	114	13	101
Age, X ± SD	43.8 ± 15.1	43.4 ± 15.6	41.5 ± 15.1
Age group			
0–19	3 (2.72)	0	3 (2.97)
20–39	51 (44.74)	7 (53.8)	44 (43.6)
40–59	38 (33.33)	3 (23.1)	35 (34.7)
60–79	22 (19.30)	3 (23.1)	19 (18.8)
Sex			
Male	65 (57.02)	5 (38.5)	60 (59.4)
Female	49 (42.98)	8 (61.5)	41 (40.6)
Underlying disease			
No	86 (75.44)	9 (69.2)	77 (76.2)
Yes	28 (24.56)	4 (30.8)	24 (23.8)
Symptom			
No	83 (72.81)	7 (53.8)	76 (75.2)
Yes	31 (27.19)	6 (46.2)	25 (24.8)
Vaccine doses			
N	114	13	101
No	6 (5.26)	0	6 (5.94)
One	1 (0.88)	0	1 (0.99)
Two	8 (7.02)	0	8 (7.92)
Three	86 (75.44)	12 (92.3)	74 (73.3)
Four	13 (11.40)	1 (7.69)	12 (11.9)
NAb ^1^			
GM ± GSD	1259.36 ± 1039.63	1742.12 ± 439.30	1207.84 ± 1052.65
Negative	4 (3.51)	0	4 (3.96)
Positive	110 (96.49)	13 (100)	97 (96.0)
IgG			
GM ± GSD	90.10 ± 40.77	84.08 ± 21.43	90.91 ± 42.98
Negative	0	0	0
Positive	114 (100)	13 (100)	101 (100)
IgM			
Negative	110 (96.49)	13 (100)	97 (96.0)
Positive	4 (3.51)	0	4 (3.96)

^1^ GM, geometric mean; GSD, geometric standard deviation; NAb, neutralizing antibody; IgG, immunoglobulin G; IgM, immunoglobulin M.

**Table 3 viruses-16-00672-t003:** Linear mixed-effects model comparison of NAbs and IgG antibodies changes at different time points in different populations ^1^.

	NAbs		IgG	
	*t* Value	*p*	*t* Value	*p*
(Intercept)	18.145	**<0.001**	23.961	**<0.001**
4 months	−3.618	**<0.001**	−3.281	**<0.01**
5 months	−4.324	**<0.001**	−3.890	**<0.001**
6 months	−2.964	**<0.001**	−5.343	**<0.001**
7 months	−2.964	**<0.01**	−6.839	**<0.001**
Healthcare workers	−1.274	0.207	−1.667	0.101
Male	1.600	0.115	1.838	0.071
Symptomatic	0.086	0.931	−0.619	0.538
Underlying diseases	−0.027	0.978	−0.536	0.594
≥35 years	−0.693	0.491	0.132	0.896

^1^ NAb, neutralizing antibody; IgG, immunoglobulin G. Values in bold indicate statistical significance (*p* < 0.05).

**Table 4 viruses-16-00672-t004:** Linear mixed-effects model comparison of various immune cells at different time points in different populations ^1^.

	CD4+ % ^2^	CD8+ %	B Cell %	NK Cell %	Total T Cell Count	Helper T Cell Count	Cytotoxic T Cell Count	B Cell Count	NK Cell Count
(Intercept)	**<0.001**	**<0.001**	**<0.001**	**<0.001**	**<0.001**	**<0.001**	**<0.001**	**<0.001**	**<0.001**
5 months	**0.028**	**0.025**	**0.006**	0.654	**0.002**	**0.014**	**0.032**	**<0.001**	**0.025**
7 months	0.140	0.071	**0.010**	0.990	0.562	0.988	0.869	0.100	0.604
Healthcare workers	0.740	0.994	**0.008**	0.388	0.913	0.358	0.844	**0.009**	0.411
Female	0.444	0.546	0.477	0.600	0.670	0.623	0.349	0.158	0.758
Symptomatic	0.774	0.378	0.547	0.079	0.235	0.457	0.237	0.415	0.347
Underlying disease	0.323	0.716	0.283	**0.048**	0.369	0.289	0.654	0.124	0.200
≥35 years	**0.020**	0.793	0.071	0.901	0.240	0.135	0.254	0.299	0.409

^1^ All numbers in the tables represent *p*-values derived from a linear mixed-effects model. Values in bold indicate statistical significance (*p* < 0.05). ^2^ Percentage: the percentage of immune cells in whole blood. Count: the absolute count of immune cells per microliter of blood.

## Data Availability

The datasets used and/or analyzed during the current study are available from the corresponding author on reasonable request.

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
