# Peer review of "Longitudinal Dynamics of Immune Response in Occupational Populations Post COVID-19 Infection in the Changning District of Shanghai, China"

_viruses, 2024, doi:10.3390/v16050672_

Round 1
Reviewer 1 Report
Comments and Suggestions for Authors
The aim of this study is to obtain more information about post-COVID immunity. By following a cohort of infected individuals for several months, the authors show evidences that the level of immunity against SARS-CoV2 is roughly maintained at least during 7 months post-infection, independently of individual parameters like sex or age. I think these data are useful, but here are some questions/remarks:
1) Most of the cohort has been infected between November 2022 and January 2023 and the study stats in March 2023, which is referred to as “month 3”. However, for some of the participants, it is actually month 6. How were the dates of diagnosis distributed?
2) What type of vaccine did the participants receive? When did they get vaccinated?
3) What were the SARS-CoV2 antigens targeted by the antibody detection kits?
4) I would put a bit more details in the legends. For example, I suppose the figures in table 4 are p-values and the ones in bold correspond to the significant ones? (By the way I would also highlight in bold the significant p-values on the graphs)
5) Figure 4: “Immune indicators in the patients during the five follow-up visits”: actually, these parameters have been quantified only at three of the visits…
It would be clearer to write “(A-F) Percentage: the percentage of immune cells… (G-K) Count: the absolute count…”
6) Line 141-142: The number of vaccinated participants is 108, but this is only among the individuals that were infected either during the 2021-2022 (13) or the 2022-2023 (95) outbreaks. The total number of vaccinated participants is 161 (108+53), which indeed gives an infection rate among vaccinated peoples of 67% (actually 67,08% and not 67,7% giving the number of infected vaccinees is 108 and not 109).
7) Line 173: “…could suggest a possible reinfection with SARS-CoV2.”

Reviewer 2 Report
Comments and Suggestions for Authors
The authors studied the dynamics of changes in the adaptive humoral response and the status of integral indices of cellular immunity (blood counts of CD4 and CD8 T cells, B cells, NK) during 3-7 months after COVID-19. The authors found a moderate downward trend in neutralizing (NAbs) and total IgG antibodies specific to SARS-CoV-2 antigens. The results clarify, but do not fundamentally change, the current understanding of the dynamics of humoral immune response parameters after COVID-19. Meanwhile, I have several criticisms regarding the content of this article:
(1). The content of the abstract may mislead the reader. It is unclear to which SARS-CoV-2 antigens the IgGs were specific; if to the nucleocapsid (N protein), then it should be explicitly stated that anti-N IgGs were determined. It should be noted that NAbs are also predominantly IgG.
(2). Materials and Methods. The principles of inclusion and exclusion in the creation of patient groups are not specified. For example, were patients with comorbid autoimmune diseases, tumor diseases, other acute and chronic infectious diseases excluded from the study? Were there any patients with proven post-COVID syndrome in the COVID-19 treated group? It is unclear which CD markers were used to verify B cells and NK due to the heterogeneity of these lymphocyte subpopulations. It is unclear to which SARS-CoV-2 antigens IgG and IgM antibodies were determined.
(3). Discussion. Limitations of the study include the fact that the authors did not study the adaptive cell-mediated response, nor did they study more private subpopulations of T cells (memory T cells, Treg, etc.) and B cells (B1, B2), which did not allow a more detailed study of the state of cellular immunity.
(4). Conclusions. "SARS-CoV-2 infection leads to induction of antibodies and activation of cellular immunity. The results obtained by the authors do not allow us to judge the activation of anti-SARS-CoV-2 cellular immunity.
Round 2
Reviewer 2 Report
Comments and Suggestions for Authors
The authors have taken the reviewer's comments into account and have been able to significantly improve the quality of their scientific article. The article can be "Accept in present form".